# An Experimental Study of Turbulent Structures in a Flat-Crested Weir-Type Fishway

**Zhiping Guo [1,2], Xihuan Sun [1] and Zhiyong Dong [3,*]**

1   College of Water Resources and Engineering, Taiyuan University of Technology, Taiyuan 030024, China; guozhiping2006@126.com (Z.G.); sunxihuan@tyut.edu.cn (X.S.)
2   Shanxi Conservancy Technical Institute, Yuncheng 044004, China
3   College of Civil Engineering and Architecture, Zhejiang University of Technology, Hangzhou 310023, China
*   Correspondence: dongzy@zjut.edu.cn; Tel.: +86-571-8529-0823



**Featured Application: This work can be used to assist to design a reasonable flat-crested weir-type fishway.**

**Abstract:** Fishways can assist fish species to overcome obstacles for performing spawning, feeding, and overwintering migrations. Flow structures in a flat-crested weir-type fishway were experimentally studied. Variations of time-averaged velocity, turbulence intensity, and Reynolds stress with longitudinal and vertical directions and flow rates were analyzed. Also, flow patterns in the longitudinal profile were given. The experiments were carried out in a large scale fishway model in the Hydraulics Laboratory at Zhejiang University of Technology. Two typical flow rates corresponding to detection and preference velocities of fish species were considered. Five different horizontal planes for each flow rate were taken. Eleven transverse lines were arranged for each horizontal plane. Ten measuring points were laid along each transverse line. Three-dimensional velocity at each measuring point was measured by acoustic Doppler velocimeter (ADV). Longitudinal and vertical time-averaged velocity distributions, longitudinal turbulence intensity distribution on the different horizontal planes, vertical turbulent intensity distribution along flow depth, and Reynolds stress distributions on the different horizontal planes and on the different cross-sections in the pool of fishway were analyzed. The experimental results showed that distribution of longitudinal velocity was characterized by topology, which constituted an apparent vertical vortex. Weir flow exhibited skimming flow in the fishway pool. Peak-value range of longitudinal turbulent intensity existed. The amplitude of variation in Reynolds stress near the surface layer reached the maximum, which provided a certain hydraulic condition for fish that favor jumping near the surface layer. This study uncovers three-dimensional flow structures, especially for turbulence characteristics, which can contribute to improving the design of crested weir fishway and to assisting fish species to pass smoothly through fishway, being of potential application value.

**Keywords:** flat-crested weir-type fishway; velocity distribution; turbulence intensity distribution; Reynolds stress distribution

## 1. Introduction

A fishway is a low-head hydraulic structure, which can assist fish species to overcome obstacles to assist in spawning, feeding, and overwintering migrations. A lot of fishways such as Doulonggang tidal sluice, Taiping sluice, and Yuxi sluice were built in sluice complexes in China during the 1960s and 1970s [1–3]. Some fishways such as Qililong fishway at Fuchunjiang hydropower station, Yangtang fishway at Hengdong pumped storage power station were built in hydropower stations [4,5].

A fishway effect is not only related to hydraulic characteristics of the pool, but also depends on swimming behavior of fish species. The investigated data [6,7] showed that fishways that fish can smoothly pass through were no more than half in the built fishways in the world. According to the UN World Water Development Report [8], freshwater fish has been reduced by 30% worldwide since the 1970s. The British fish monitoring data indicated that the regression rate of Atlantic salmon apparently lowered in the past decades [9], and the effectiveness of fishway was doubtful. So it is of important practical significance to further study the hydraulic characteristics of fishway based on the swimming behavior of fish species, especially in the effect of turbulent structures in fishway on fish swimming behavior.

Fishway can be divided into pool-type [10–12], Denil-type [13], and nature-like bypass-type [14–17] though other types exist. Designs of almost all types of fishways are based on the dissipation principle so as to lower flow velocity for facilitating anadromous fish species. The basic principle of pool-type fishway consists of a series of pools separated by a certain number of baffles, which breaks up the total head into a series of small heads to lower the velocity of fish passage and create a waterway that fish can swim through. The baffles can be used to control the flow pattern in the pool and provide an environment that can be of both swimming and rest for fish. In general, according to the shape of baffle, pool-type fishway can be divided into vertical slot, crested weir, orifice, and their combined fishways. The crested weir fishway is suitable for fish behavior that has a preference for swimming and jumping in the surface layer. Up to now, hydraulic design of most of fishways focuses only on mean velocity, neglecting turbulent characteristics in fishways. Fish can detect turbulence by their inner ear, lateral line, and nerve hillock [18]. However, how do fish cope with external stimulation? It has been not reported yet. Some studies on the effect of turbulence on fish behavior, energy, and distribution showed that fish can benefit from some turbulent regimes, for example, the experimental results of Liao, Beal, and Lauder et al. [19] showed that adult rainbow salmon could benefit from the vortex energy when swimming in steady vortices behind an obstacle. The salmon swam in the form of ski-jump among the vortices, which could reduce the axial muscle activity. It means that the energy cost of a fish swimming in turbulent flow could be reduced if the fish swam appropriately. The similar observed results showed that adult sockeye salmon utilized vortex zones to minimize its energy cost on the way of anadromous migration [20,21]. Herskin and Steffensen's [22] experiment revealed that fish could benefit from a vortex due to the leading fish in the fish school, for example, the beating frequency of their tail fin could lower 9–14%, and the energy cost could reduce 9–23% in the rear of a sea bass school than that in the front. If flow in a fishway would be of such ecology-friendly flow structures, it could greatly contribute to fish swimming. It can be said that the experimental study of flow characteristics in a fishway pool focused mainly on a one-dimensional aspect because of the limit of test means in the past. Actually, fish species are in three-dimensional flow fields, and experience effects of three-dimensional turbulent structures. The complicated three-dimensional turbulent structures have been little known, for example, the work of Pisaturo et al. [23] underlined the necessity to study the fish habitat with a 3D hydraulic simulation approach. In this paper, the three-dimensional instantaneous velocity at each measuring point in a fishway pool was measured by acoustic Doppler velocimeter (ADV) in a large-scale pool-weir fishway flume, and turbulent structures were analyzed in detail.

## 2. Experimental Facility and Methodology

The experimentation was conducted in a large-scale fishway flume in the Hydraulics Laboratory at the Zhejiang University of Technology. The experimental setup of fishway flume is shown in Figure 1, which consists mainly of inflow, working, and outflow sections. The flume was 20.0 m long, 0.6 m wide, and 1.0 m deep. The two sides were made of tempered glass. The working section consisted of 4 pools separated by baffles, each pool was 0.9 m long, and the length to width ratio of pool was 1.5. The baffle material was polyethylene plate. Baffle height over each step was 0.65 m. Step heights for 4 pools were 0.20, 0.15, 0.10, and 0.05 m, respectively. The flow rate was measured by a rectangular sharp-crested weir at the end of the flume, and the water level was measured by an automatic water level measuring

system. Two flow rates $Q_1$ = 20.79 L/s and $Q_2$ = 30.04 L/s were selected, the corresponding weir flow velocity were $U_1$ = 49.49 cm/s and $U_2$ = 62.50 cm/s, respectively. Three-dimensional velocities at each measuring point were measured with SonTek Micro ADV.

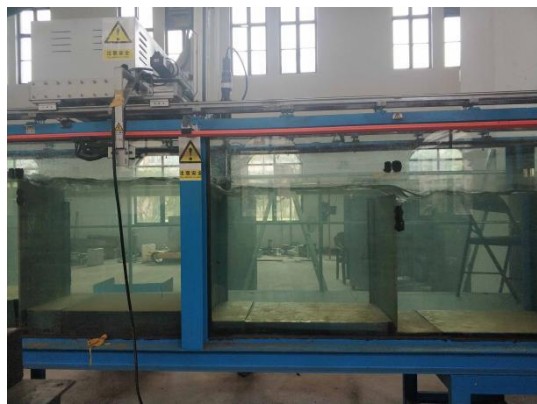

**Figure 1.** Setup of the flat-crested weir-type fishway flume.

The coordinates of the fishway pool can be defined as $x$, $y$, and $z$ axes along longitudinal, transverse, and vertical directions as shown in Figure 2, respectively. The origin is located at the left lower corner of intersection between $x$ and $y$ axes. In the experiment, 2 flow rates were selected, 5 different planes were taken for each flow rate, 11 transverse lines ($x$ = 5, 10, 15, 20, 25, 35, 45, 55, 65, 75, and 85 cm) were measured on each horizontal plane, and 10 measuring points with spacing 5cm were arranged at each transverse line as shown in Figure 3, respectively. Based on the ADV measurements and the theory of turbulent flow, variations in longitudinal velocity along flow direction and depth, distribution of vertical velocity along depth, characteristics of vertical vortex, longitudinal and vertical turbulence intensity distributions, and Reynolds stress distribution were analyzed.

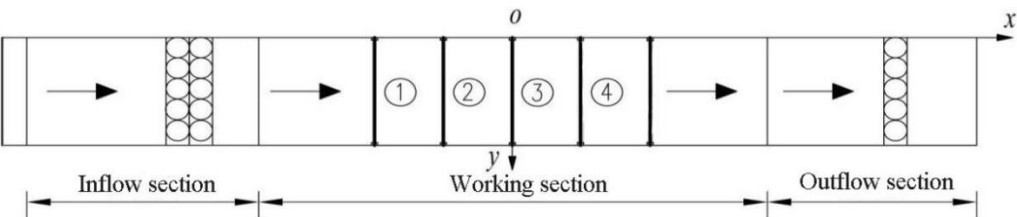

**Figure 2.** Sketch and coordinates definition of the flat-crested weir-type fishway flume.

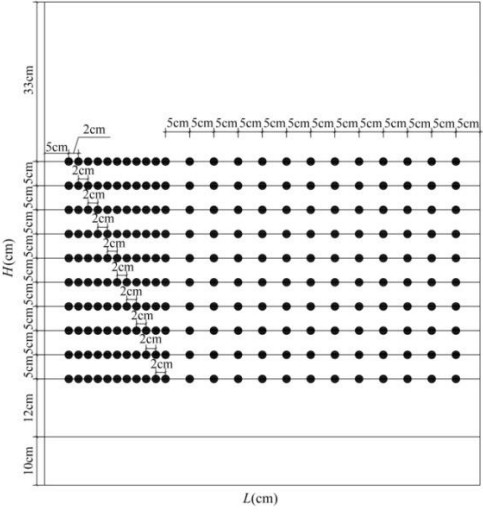

**Figure 3.** Layout of measuring points.

## 3. Results and Discussions

### 3.1. Time-Averaged Velocity Distribution

#### 3.1.1. Variation in Longitudinal Time-Averaged Velocity along Flow Direction

When flow rates $Q_1 = 20.79$ L/s and $Q_2 = 30.04$ L/s, variation in longitudinal velocity $u$ along flow direction in the fishway pool is shown in Figure 3. It follows from the figure that flow profiles along each depth are characterized by topology, constituting considerable vertical vortex, the velocity in the center of the vortex is almost zero. It is found from a comparison between Figure 4a,b that the vertical vortex gets flattened with an increase in flow rate, and the reason is that the pool depth plunged by weir flow deepens due to an increase in flow rate, thus delaying formation of the surface skimming flow.

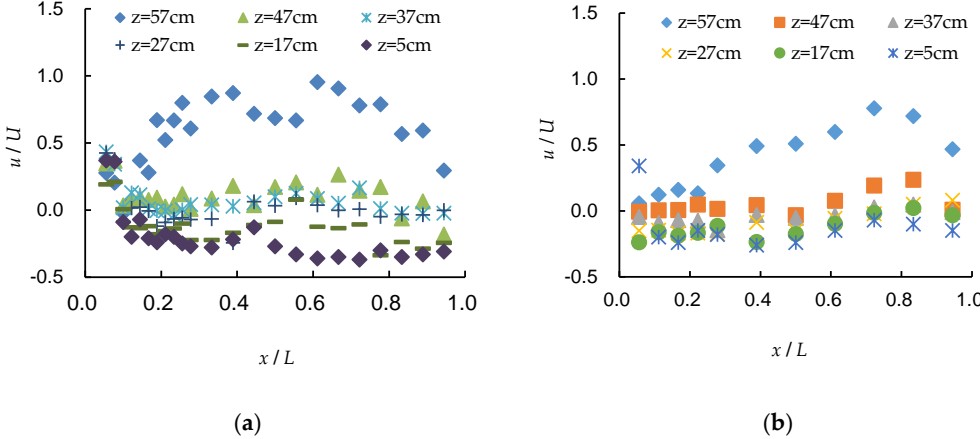

**Figure 4.** Variation in longitudinal velocity along the flow direction. (**a**) $Q_1 = 20.79$ L/s; (**b**) $Q_2 = 30.04$ L/s.

#### 3.1.2. Distribution of Longitudinal Time-Averaged Velocity along Depth

Distribution of longitudinal time-averaged velocity $u$ along the pool depth is shown in Figure 5. It can be seen that, under flow rates $Q_1 = 20.79$ L/s and $Q_2 = 30.04$ L/s, there exists a certain similarity in longitudinal velocity distribution except close to the baffle, and the maximum velocity at each cross-section is located in the free surface and then gradually decreases with an increase in pool depth.

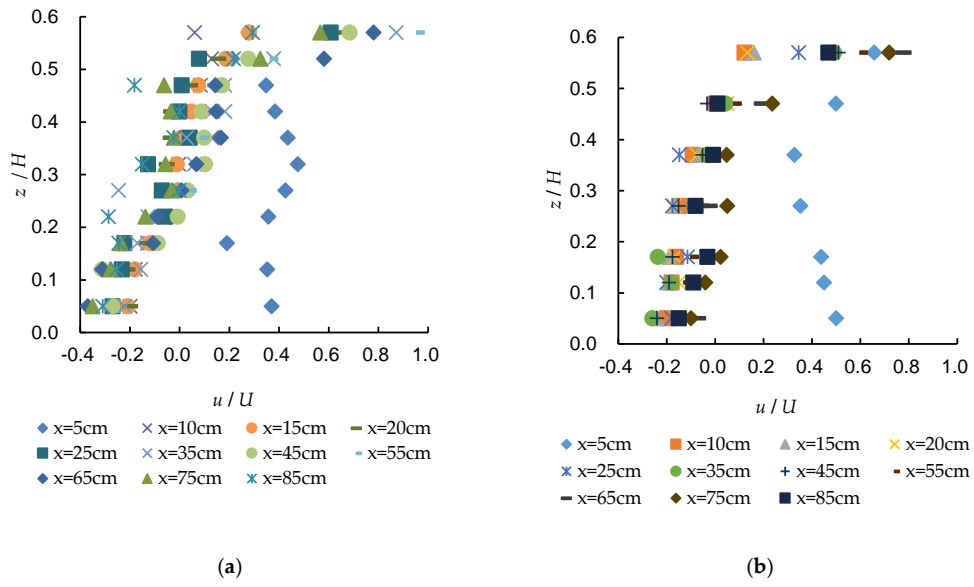

**Figure 5.** Longitudinal velocity distribution along flow depth. (**a**) $Q_1 = 20.79$ L/s; (**b**) $Q_2 = 30.04$ L/s.

### 3.1.3. Distribution of Vertical Time-Averaged Velocity along Pool Depth

In the fishway pool, distribution of vertical velocity $w$ along depth is shown in Figure 6a,b for flow rates $Q_1 = 20.79$ L/s and $Q_2 = 30.04$ L/s. We can see from the figures that the closer to the impingement region due to weir flow, the higher the vertical velocity. At the same cross-section, distribution of vertical velocity along pool depth is basically uniform, so the flat-crested weir flow can be considered as a vertical two-dimensional flow.

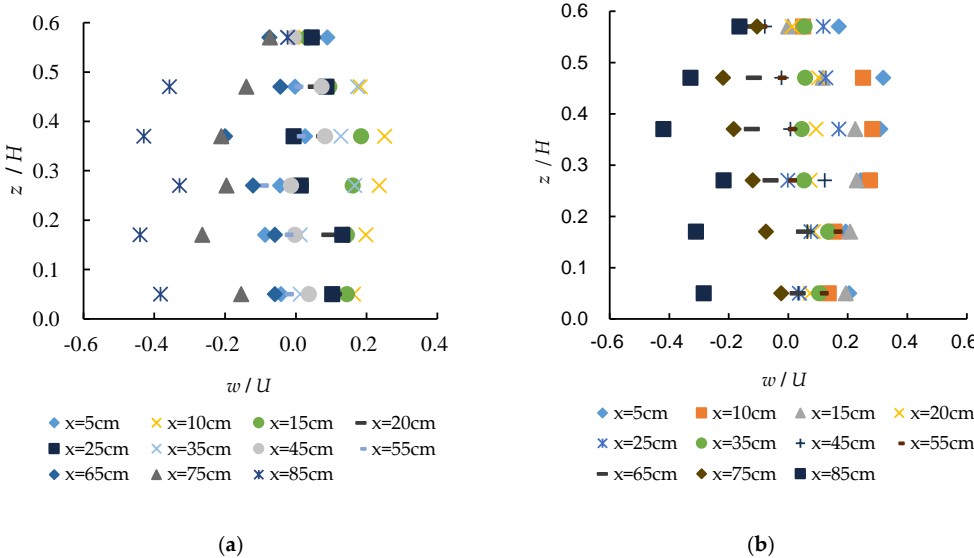

(**a**)　　　　　　　　　　　　　　　　　　　　　　　　　(**b**)

**Figure 6.** Vertical velocity distribution along flow depth. (**a**) $Q_1 = 20.79$ L/s; (**b**) $Q_2 = 30.04$ L/s.

### 3.2. Vertical Vortex in the Pool

Figure 7 shows the measured velocity vector chart in a longitudinal profile for the two flow rates. In the figure, the abscissa $L$ denotes the pool length, and the ordinate $H$ the pool depth. It follows from Figure 7 that after weir flow plunged into the pool, it flows in the form of a skimming flow along surface layer. Because of obstruction due to downstream baffle, one part of the skimming flow discharges into the downstream pool, and the other part forms a vertical vortex in the longitudinal profile. The vertical vortex is characterized by a larger outer velocity vector, and close to the vortex centre by a smaller velocity vector. This flow pattern can contribute to fish rest during swimming.

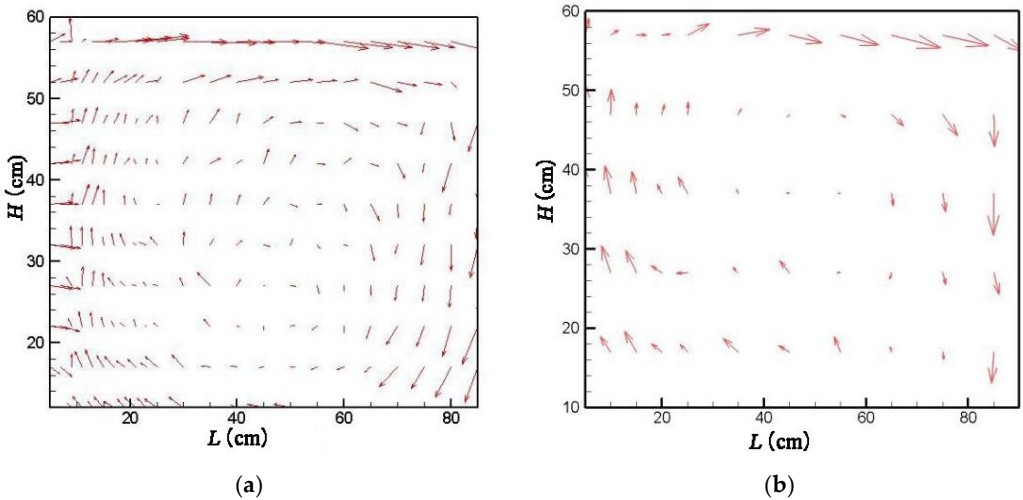

(**a**)　　　　　　　　　　　　　　　　　　　　　　　　　(**b**)

**Figure 7.** Velocity vectors on the *xoz* plane. (**a**) $Q_1 = 20.79$ L/s; (**b**) $Q_2 = 30.04$ L/s.

### 3.3. Turbulence Intensity Distribution

Longitudinal and vertical turbulence intensities can be respectively expressed as

$$Tu = \frac{\sqrt{u'^2}}{U}, Tw = \frac{\sqrt{w'^2}}{U}, \tag{1}$$

where $U$ denotes the weir flow velocity.

Figure 8 shows the variation in longitudinal turbulence intensity for the two flow rates. It follows from the figure that turbulence intensity in surface layer ($z = 57$ cm) is apparently greater than that in the lower layer, it reaches the maximum value in the impingement region due to weir flow, that is, there exists a peak value range of turbulence intensity, and then decreases along flow direction. Variation in the range of turbulence intensity in the lower layer flow is smaller. Because the zone close to upstream baffle is in the impingement region, turbulent mixing is intense, free surface fluctuates intensely, and the flow abruptly turns into longitudinal flow from vertical flow. The larger the flow rate, the more intense the fluctuation in surface layer flow, and the larger the variation in range of turbulence intensity.

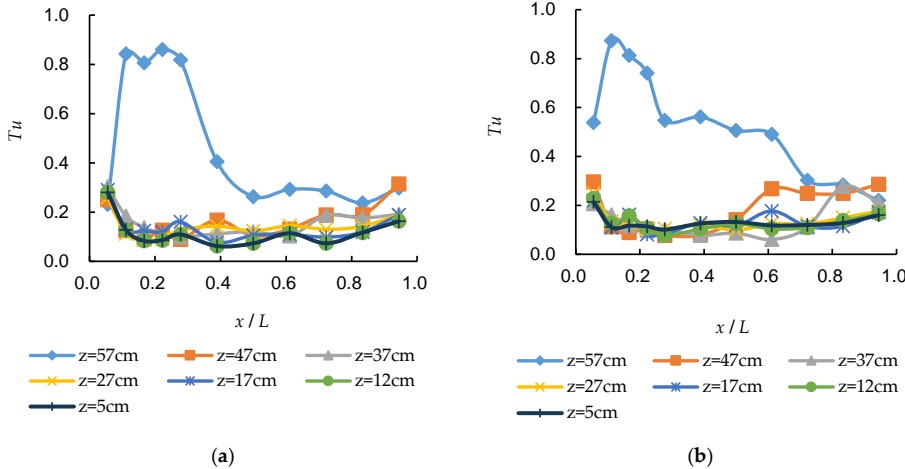

**Figure 8.** Longitudinal turbulence intensity distribution. (**a**) $Q_1 = 20.79$ L/s; (**b**) $Q_2 = 30.04$ L/s.

Variation in vertical turbulence intensity along the pool depth at each cross-section for two flow rates is shown in Figure 9. It can be seen through the comparison between the two flow rates that in the impingement region of weir flow ($x = 10, 15,$ and $20$ cm), turbulence intensity is larger in the surface layer, however, it gradually decreases and keeps stable with an increase in depth. The reason is that velocity fluctuates intensely due to turbulent diffusion of impinging jet in the pool.

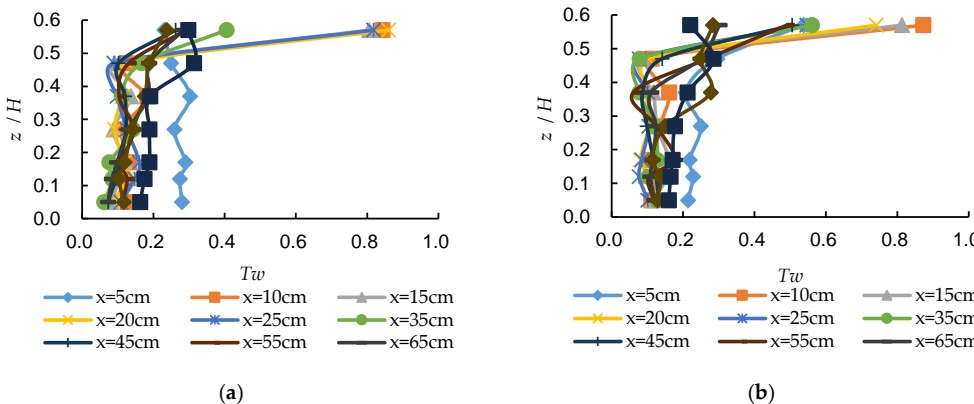

**Figure 9.** Vertical turbulence intensity distribution. (**a**) $Q_1 = 20.79$ L/s; (**b**) $Q_2 = 30.04$ L/s.

### 3.4. Reynolds Stress Distribution

On the *xoy* and *xoz* planes, Reynolds stress can be respectively expressed as follows:

$$\eta_{xy} = -\frac{\overline{u'v'}}{U^2}, \eta_{xz} = -\frac{\overline{u'w'}}{U^2}. \tag{2}$$

Reynolds stress equation can be written as:

$$\underbrace{\frac{\partial}{\partial t}\left(\overline{V'_i V'_j}\right)}_{changing\ rate} + \underbrace{\overline{V_l}\frac{\partial \overline{V'_i V'_j}}{\partial x_l}}_{advection} = \underbrace{\frac{\partial}{\partial x_l}\left[-\overline{V'_i V'_j V'_l} - \frac{\overline{p'}}{\rho}\left(\delta_{jl}V'_i + \delta_{il}V'_j\right) + v\frac{\overline{\partial V'_i V'_j}}{\partial x_l}\right]}_{diffusion},$$

$$\underbrace{-\left(\overline{V'_i V'_l}\frac{\partial V_j}{\partial x_l} + \overline{V'_j V'_l}\frac{\partial V_i}{\partial x_l}\right)}_{production} - \underbrace{2v\frac{\overline{\partial V'_i}\frac{\partial V'_j}{\partial x_l}}{\partial x_l}}_{dissipation} + \underbrace{\frac{\overline{p'}}{\rho}\left(\frac{\partial V'_i}{\partial x_j} + \frac{\partial V'_j}{\partial x_i}\right)}_{stress-strain}. \tag{3}$$

It follows from the Reynolds stress equation that variation in Reynolds stress depends upon comprehensive effects of diffusion, production, dissipation, and stress–strain of turbulent flow. On the *xoy* plane, variation in Reynolds stress along flow direction is shown in Figure 10. We can see from the figure that amplitude of variation in Reynolds stress reaches the maximum in the surface layer, which provides a hydraulic condition for fish having a preference for jumping in the free surface. The amplitude of variation in Reynolds stress is smaller below the free surface, however, it increases with an increase in flow rate. In the impingement region of weir flow, Reynolds stress fluctuates intensively, and it was found combining with vertical turbulence intensity (shown in Figure 8) that the range that turbulence intensity was larger is the same as Reynolds stress.

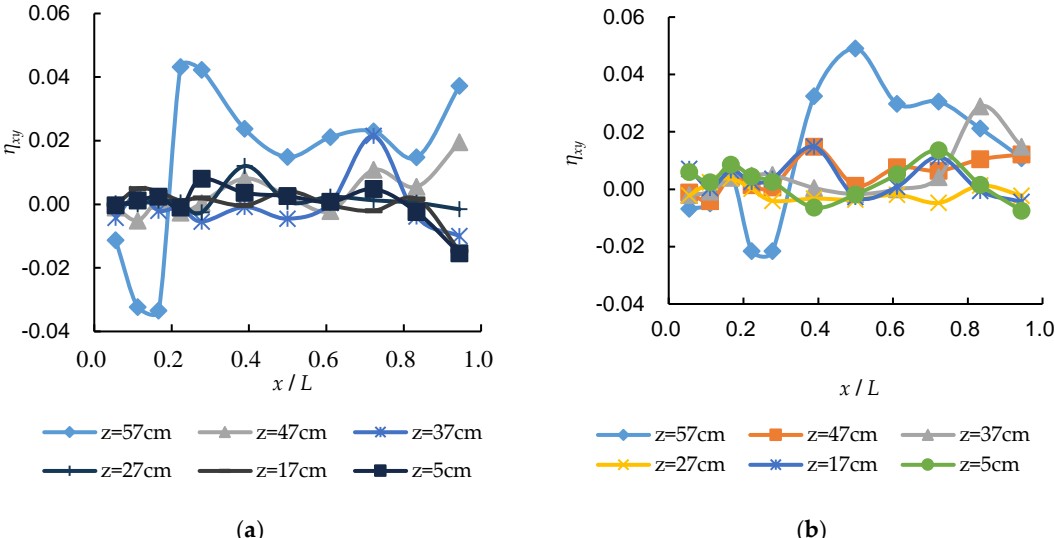

**Figure 10.** Variation in Reynolds stress along flow direction on the *xoy* plane. (**a**) $Q_1$ = 20.79 L/s; (**b**) $Q_2$ = 30.04 L/s.

Vertical distribution of Reynolds stress on the plane *xoz* is shown in Figure 11. It follows from Figure 11 that Reynolds stress fluctuates intensively close to upstream baffle (impingement region of weir flow) and downstream baffle, which resulted from the formation of a vertical vortex after weir flow plunged into the fishway pool, and the existence of ascending and descending flows close to the upstream and downstream baffles.

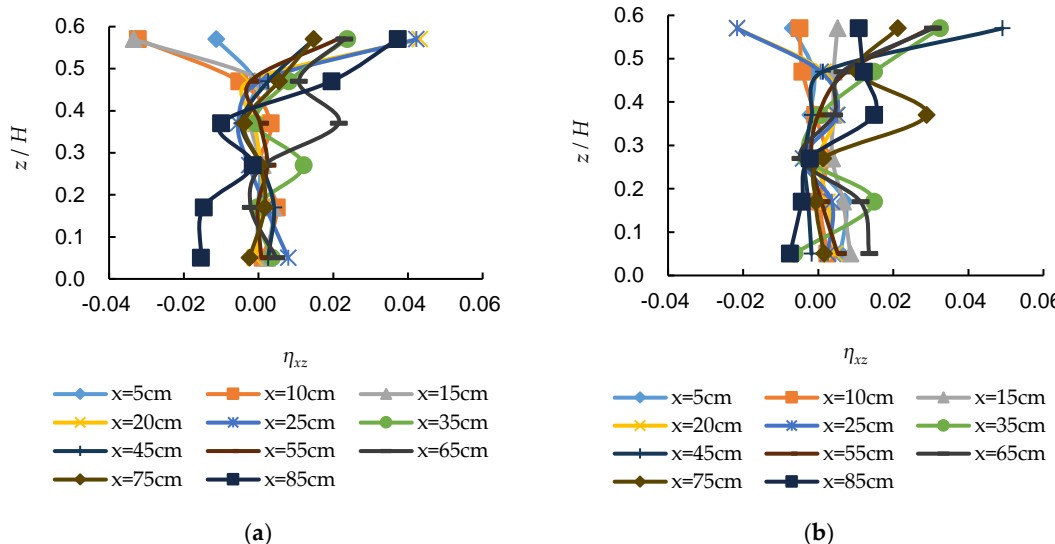

**Figure 11.** Vertical distribution of Reynolds stress on the *xoz* plane. (**a**) $Q_1 = 20.79$ L/s; (**b**) $Q_2 = 30.04$ L/s.

## 4. Conclusions

Through the experimental study of turbulent structures in the flat-crested weir-type fishway, we can draw some conclusions as follows:

Distribution of longitudinal time-averaged velocity was characterized by topology, which constituted considerable vertical vortex. The vertical vortex got prolate when flow rate increased, and impingement region of weir flow deepened to delay formation of surface skimming flow. Vertical time-averaged velocity close to the impingement region of upstream baffle being higher, and its distribution along the pool depth was basically uniform at the same cross-section. Weir flow exhibited a skimming flow in the fishway pool. One part of the skimming flow discharged into the downstream pool, and the other part formed a vertical vortex on the longitudinal profile in the pool because of a block of the downstream baffle, which contributed to fish rest during anadromous migration. Longitudinal turbulence intensity in the surface layer was larger than that in the lower layer, and reached the maximum value in impingement region of weir flow. Peak-value range of longitudinal turbulence intensity existed and then reduced along flow direction. Vertical turbulence intensity was larger in the surface layer of impingement region and decreased with an increase in the depth. The amplitude of variation in Reynolds stress in surface layer reached the maximum, which provided a hydraulic condition for fish that favor jumping near the surface layer. The amplitude of variation below surface layer was smaller, but it enlarged with an increase in flow rate.

In the past, the hydraulic design of most of fishways focused merely on mean velocity due to a limit of test means and ignoring turbulence characteristics in fishways. However, the engineering practice showed that many fishways failed to allow fish species to pass. Actually, fish species are in the three-dimensional flow fields, the flow structures are complicated, including interaction among different turbulent jets such as free jet, wall jet, impinging jet, and multi-jet in a confined space, also accompanying intense horizontal and vertical vortices. How do fish species to react to the complicated turbulent structures? It is valuable to conduct fish tests based on the experimental study of turbulent structures, and to develop the relation between fish species and the turbulent structures.

**Author Contributions:** Introduction, Z.G. and Z.D.; experimental facility and methodology, Z.G. and Z.D.; time-averaged velocity distribution, Z.G. and X.S.; vertical vortex in the pool, X.S. and Z.G.; figures and tables, Z.G. and X.S.; data curation, Z.G., Z.D., and X.S.; writing—original draft preparation, Z.G. and Z.D.; writing—review and editing, Z.G. and X.S.

**Funding:** This study was funded by the National Natural Science Foundation of China (NSFC), Grant Number 51779225.

**Conflicts of Interest:** The authors declare no conflict of interest.

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
