# Peer review of "An Experimental Study of Turbulent Structures in a Flat-Crested Weir-Type Fishway"

_applsci, doi:10.3390/app9194040_

Round 1

Reviewer 1 Report

Minor editing is required to improve the quality of the English. Otherwise a sounds report.

Reviewer 2 Report

Did the laboratory speed measurements with ADV take into account the effects of the walls?

Where were the measuring points in the verticals?

What were the declines  of the flat-crested weir-type fishway flume

For which fish species were the tests conducted? Such information (from an ichthyologist) is important to avoid design and construction errors

What location of the fish pass would the authors suggest? This is important because of the fish's ability to find a fish pass

What would the water depth be at the bottom stand? Are the current attraction characteristics taken into account?

For which no equation na was used to determine the stress value using three velocity components (the equation is presented in the publication Tymiński, T. and Mumot, J., 2015.)

Was the study preceded by a local visit?

The chamber indicates strong vortices that can prevent fish from flowing through the chamber, or the authors can indicate places where the fish can wait

Extend the discussion with publications including:

Teppel, A. and Tyminski, T., 2013. Hydraulic research for successful fish migration improvement–” nature-like” fishways. Civil and Environmental Engineering Reports, (10), pp.125-137.

TymiÅ„ski, T. and KaÅ‚uża, T., 2013. Effect of vegetation on flow conditions in the “Nature-Like” fishways. Rocznik Ochrona Åšrodowiska, 15(cz. 1), pp.348-360.

Bartnik W., Książek L., Florek J., WyrÄ™bek M., Leja M., 2012 Analysis  of  hydrodynamic  bed  stability  conditions  in  rock  fishways, Acta Scientiarum Polonorum, Seria Formatio Circumiectus, 11 (2), 3-14

Bartnik W., Książek L., WyrÄ™bek M., StrutyÅ„ski M., 2012, Bedload  equalibrium  measurementsalong canoe-fishways, Acta Scientiarum Polonorum, Seria Formatio Circumiectus, 11 (4), 5-16

Hämmerling, M., KaÅ‚uża, T., & Walczak, N. (2017). Hydraulic conditions of water flow in seminatural fish pass, A case study of the Skórka barrage on the GÅ‚omia river. Acta Scientiarum Polonorum. Formatio Circumiectus, 16(2), 85.

Hämmerling, M., & KaÅ‚uża, T. (2018). Analysis of Fish Migration Potential Through the Seminatural Fish Pass on an Example the Skórka Barrage on the GÅ‚omia River. Rocznik Ochrona Åšrodowiska, 20.

Reviewer 3 Report

The article focuses on the study of turbulence in weir weir-type fishway.

The article is interesting and the topic important to design fish passes.

I have some suggestions and corrections:

line 51. Insert references. line 77. you can add the following sentence. "For example the work of Pisaturo et al. 2017 underlines the necessity to study the fish habitat with a 3D hydraulic simulation approach."
Reference: Pisaturo, G.R., Righetti, M., Dumbser, M., Noack, M., Schneider, M., Cavedon, V., 2017. The role of 3D-hydraulics in habitat modelling of hydropeaking events. Sci. Total Environ.575, 219230. https://doi.org/10.1016/j.scitotenv.2016.10.046 lines 99-100. You talk about some depths. Which are these depths? line 105. Dot at the end of the sentence figure 2. Better draw also where are the measuring points. Moreover, which is the y plane that you considered in the study? From the figure it seems the plane near a wall. line 126. I think that you can't compare this results with open channel flow because here you have a vertical circulation. Better compare with cavity flow. Insert reference and check if your results are comparable with cavity flow theory. figure 5. Put 0 value in the x axis figures 7-10 are difficult to read. Can you improve?

General comment. In my opinion you have to improve the quality of the paper, improving the methods, and the discussion of the results. The results are interesting but not well explained. 

Round 2

Reviewer 3 Report

Ok, thanks for the corrections. Now the quality of your work increased.
